# Mid-Term Outcome of Ventricular Arrhythmias Catheter Ablation in Patients with Chronic Coronary Total Occlusion Compared to Ischemic and Non-Ischemic Patients

**DOI:** 10.3390/jcm11237181

**Published:** 2022-12-02

**Authors:** Maria Lucia Narducci, Giampaolo Niccoli, Francesco Flore, Francesco Perna, Gianluigi Bencardino, Rocco Antonio Montone, Gemma Pelargonio, Filippo Crea

**Affiliations:** 1Dipartimento di Scienze Cardiovascolari, Fondazione Policlinico Universitario Agostino Gemelli IRCCS, 00168 Rome, Italy; 2Dipartimento di Medicina e Chirurgia, Università di Parma, 43125 Parma, Italy; 3Istituto di Cardiologia Università Cattolica del Sacro Cuore Roma, Department of Cardiovascular Sciences, 00168 Rome, Italy

**Keywords:** personalized medicine, ventricular arrhythmias, radiofrequency catheter ablation, coronary chronic total occlusion, therapeutic strategy

## Abstract

Chronic coronary total occlusions (CTO) are considered an emerging predictor of ventricular arrhythmias (VAs), but currently there are few data on arrhythmic outcomes in patients affected by CTO undergoing radiofrequency catheter ablation of VAs. This study sought to evaluate the impact of unrevascularized CTO on the recurrence of VAs after catheter ablation. This was a single-center retrospective study enrolling 120 patients between 2015 and 2020. All patients were admitted for ventricular tachycardia (VT) or high premature ventricular contractions burden (>25% detected by Holter ECG), without evidence of acute coronary syndrome; they underwent coronary angiography, electrophysiology (EP) study, and three-dimensional electroanatomic mapping (3D-EAM) followed by VAs ablation. Twenty-eight patients (23%) of 120 patients showed CTO at coronary angiography. At baseline, the CTO group presented with higher prevalence of hypertension, chronic renal disease, systolic ventricular dysfunction, secondary prevention ICD implantation, and higher rate of LAVA by 3D-EAM compared with the non-CTO group. At a median follow-up of 15 months (range 1–96 months) after catheter ablation, the only independent predictor of VAs recurrence was the presence of moderate to severe left ventricular (LV) dysfunction. Therefore, the presence of CTO does not predict VAs recurrence after catheter ablation, which is instead predicted by LV dysfunction.

## 1. Introduction

Sustained ventricular arrhythmias (VAs) might be both a manifestation of an underlying heart disease and an idiopathic condition affecting apparently healthy hearts. In the setting of ischemic heart disease, management of sustained VAs is a topic of growing interest and catheter ablation has been demonstrated to be a valid therapeutic strategy [1].

Chronic coronary total occlusion (CTO) is a common condition involving 18% to 52% of patients with coronary artery disease, and several studies have suggested their role in VAs recurrence among patients with chronic coronary syndrome [2,3]. In particular, among CTO patients with ICD implanted for primary and secondary prevention of SCD, the rate of appropriate ICD therapies can be as high as 25% and 80%, respectively, at mid-term follow up [2,3,4]. Despite this, only a few studies have investigated the role of CTOs on VAs recurrence rate in patients undergoing radiofrequency catheter ablation [5,6].

In our retrospective study, we aimed to investigate a cohort of patients admitted for VAs and treated by radiofrequency catheter ablation, in order to identify predictors of VAs recurrence in unrevascularized CTO versus non-CTO patients.

## 2. Materials and Methods

### 2.1. Study Design and Patient Selection

In this retrospective observational study, we consecutively enrolled 120 patients referring for VAs ablation at our Department of Cardiovascular Science between 2015 and 2020.

VAs were defined as:−High burden of premature ventricular contractions (PVCs) (≥25% of total beats in 24 h ECG Holter monitoring) or non-sustained ventricular tachycardia (NSVT) during 24 h ECG Holter monitoring or exercise maximal stress test;−Sustained ventricular tachycardia (VT) or ventricular fibrillation (VF).

Patients enrolled in the study underwent the same clinical and instrumental workup, including careful medical history and physical examination, 2D-transthoracic echocardiography, coronary angiography, 3D-electroanatomic mapping (3D-EAM), and VAs radiofrequency catheter ablation. All patients gave informed medical consent for the study. The study was approved by the local Ethical Committee. The study complied with the Declaration of Helsinki as revised in 2013.

Left and right ventricular global systolic function and LV regional function were evaluated in accordance with EACVI/ASE recommendations for echocardiography [7].

At coronary angiography, the existence of CTO was defined as 100% occlusion (TIMI 0 flow) of a coronary artery for a duration greater than or equal to 3 months [8]. Coronary revascularization during hospitalization and acute coronary syndrome were exclusion criteria.

Ventricular 3D-EAM and the consequent radiofrequency catheter ablation were performed according to the ECG exit site of clinical ventricular tachycardia or monomorphic PVCs. In one third of patients, antiarrhythmic drugs were discontinued before the electrophysiological study in order to facilitate VAs induction during the electrophysiological study. Patients with an indication of ICD as recommended by current guidelines were finally implanted [9,10].

### 2.2. 3D-Electroanatomic Ventricular Mapping

Three-dimension electroanatomical mapping (3D-EAM) of the left and right ventricle with the CARTO system (Biosense Webster, Diamond Bar, CA, USA) or the NavX Ensite Velocity (St. Jude MedicalTMInc., Milwaukee, WI, USA) was also performed, using a diagnostic catheter (Pentaray, Biosense Webster, or HD grid St. Jude MedicalTM). For both ventricles a value of >1.5 mV defined normal endocardial bipolar electrogram amplitude, and a value < 0.5 mV defined endocardial “bipolar scar”. We evaluated bipolar scar localization and area measurement (cm^2^, % of total mapped area). A value < 8 mV defined left ventricle (LV) endocardial “unipolar scar”, and a value < 5 mV defined right ventricle endocardial unipolar scar. We evaluated unipolar scar localization and area measurement (cm^2^, % of total mapped area). The endocardial unipolar electrogram recording information may provide a valuable clue suggesting the presence of a transmural electroanatomic scar, as previously reported [11,12,13,14]. 

Baseline local abnormal ventricular activities (LAVA) were defined as sharp high-frequency ventricular potentials, possibly of low amplitude, distinct from the far-field ventricular electrogram occurring anytime during or after the far-field ventricular electrogram in sinus rhythm or before the far-field ventricular electrogram during VT that displayed fractionation or double or multiple components separated by very-low-amplitude signals or an isoelectric interval and were poorly coupled to the rest of the myocardium. These high-frequency sharp signals were considered indicative of local electric activity arising from pathological tissue. 

### 2.3. Ventricular Arrhythmias Catheter Ablation

All ablation procedures were performed under 12-lead ECG, oximetry, and left radial artery blood pressure monitoring. Catheter ablation was performed by a 3.5 mm distal tip irrigated catheter (Navistar Thermocool or Navistar Smarttouch, Biosense Webster, orFexAbility, St. Jude MedicalTM). 

For right ventricular VAs, an ablation catheter was advanced into the right ventricle by venous femoral access under a fluoroscopic guide. For left ventricular VAs, either transseptal or aortic retrograde approaches were utilized with multipolar mapping and catheter ablation (BW), as indicated below. The latter approach was chosen in case of ECG suspicion of aortic cusps, mitral-aortic continuity, or left ventricular outflow tract (LVOT) origin of VAs. Intravenous heparin was given with a target-activated clotting time of 300 s, in the case of left ventricular 3D-EAM. Epicardial access was based on clinical data and endocardial electroanatomic mapping data.

Identification of the original site of arrhythmia was performed by activation mapping in a patient with hemodynamically well-tolerated VAs or with a high burden of monomorphic PVCs. In patients presenting with a high burden of monomorphic PVCs, the earliest site of activation during activation mapping was confirmed by 12D-ECG pace-mapping, representing the endpoint of ablation. 

In patients with hemodynamically well-tolerated sustained monomorphic VAs, the critical isthmus of clinical VT identified by activation mapping was the endpoint of ablation. In patients with unstable VT, the suppression of all low abnormal voltage activities (LAVA) was the endpoint. 

Radiofrequency was acted under power control, with a power range of 35–50 watts and a maximum temperature of 43° degrees, with irrigated catheter ablator. In case of ineffectiveness of the ablative procedure, or if VAs recurred after interruption of the radiofrequency, the arrhythmia activation mapping was re-evaluated and eventually repeated. In case of arrhythmia disappearance, fifteen minutes were expected to unmask possible effects of the peri-wound oedema induced by radiofrequency. Complete LAVA abolition was confirmed by detailed re-mapping after ablation. The acute success after catheter ablation was defined as a composite endpoint of non-VAs inducibility + complete LAVA abolition + acute PVC suppression. The protocol for VT induction included triple ventricular extra-stimuli performed in two right ventricular sites (respectively right ventricular apex and outflow tract). 

In order to rule out possible complications, the patient underwent clinical and instrumental monitoring, by ECG and transthoracic echocardiogram, for at least 24 h after the procedure.

### 2.4. Follow-Up

Mid-term follow-up included clinical evaluation and 24 h ECG Holter monitoring or ICD interrogation, performed every 6 months from discharge. VAs recurrence indicated as monomorphic PVCs ≥ 25% at 24h ECG Holter or SVT or VF or appropriate ICD interventions were the composite endpoint of our study.

### 2.5. Statistical Analysis

Continuous data were expressed as mean ± standard deviation and were compared using a Student t-test. Frequencies were compared using the chi-square test.

To determine whether baseline variables were independently associated with arrhythmic events at follow-up, a Cox proportional-hazards regression model was applied. The results of the Cox regression analysis are represented as hazard ratio (HR) and 95% confidence intervals (CI). The event-free survival curve was plotted using the Kaplan–Meier method with the statistical significance examined by the Mantel-Haenszel (log-rank test). The level of statistical significance was set at a 2-tailed alpha level < 0.05. All statistical analyses were performed with SPSS^®^ version 20.0 software (© Copyright IBM Corporation 1994, 2017).

## 3. Results

### 3.1. Baseline Findings

One-hundred and twenty consecutive patients presenting with VAs were treated with radiofrequency catheter ablation and characterized by heterogeneous underlying heart diseases. As summarized in Figure 1, 42 (35%) patients had ischemic heart disease, 33 patients (27%) presented an apparently healthy heart (with idiopathic outflow tract VT), 16 patients (13%) and 15 patients (12%) had a diagnosis of post-inflammatory cardiomyopathy or idiopathic dilated cardiomyopathy, respectively, 4 patients (3%) had arrhythmogenic cardiomyopathy, 3 patients (2.5%) hypertrophic cardiomyopathy, 4 patients (3%) grown up-congenital heart disease, and 4 patients (3%) valvular heart disease. 

In the ischemic heart disease group, we identified a subset of 28 subjects exhibiting unrevascularized CTO. As summarized in Figure 2, among these patients, right coronary artery (RCA) was more frequently involved (15 patients with RCA CTO, 54%) than left anterior descending artery (LAD) (8 patients, 28%) and left circumflex artery (LCA) (5 patients, 18%). In all 8 patients with LAD CTO, the latter was the infarct-related artery (IRA) CTO; among non-LAD CTO patients, CTO was the IRA in 14 patients (RCA CTO in 11 and LCA CTO in 3). Table 1 summarizes clinical and diagnostic baseline findings between CTO and non-CTO groups.

The CTO group exhibited higher age (mean age 68 vs. 56 y/o, *p* < 0.001), higher male gender (96% vs. 68%, *p* = 0.002), hypertension (86% vs. 47%, *p* < 0.001), and chronic kidney disease (57% vs. 27%, *p* = 0.02) prevalence as compared with the non-CTO group. The CTO group compared with the non-CTO group exhibited lower LVEF (36% vs. 48%, *p* < 0.001), greater prevalence of moderate to severe LV dysfunction (64% vs. 30%, *p* = 0.006) and of akinetic ventricular areas (75% vs. 22%, *p* < 0.001), as shown in Table 2. Accordingly, in the CTO group the prevalence of ICD carriers was higher than in the non-CTO group (82% vs. 44%, *p* = 0.005), as well as the prevalence of sustained VAs on admission (96% versus 53%, *p* < 0.001) and of antiarrhythmic therapy with amiodarone (42% vs. 19%, *p* = 0.01). In the subgroup of ischemic patients (43 patients, 28 with diagnosis of CTO and 15 without CTO), the CTO group presented more frequently with previous myocardial infarction (27/28 vs. 9/15 *p* = 0.02) and with LVEF < 35% (16/28 vs. 4/15, *p* = 0.03).

### 3.2. 3D Electroanatomic Mapping and Catheter Ablation Findings

In the CTO group, on the basis of VAs ECG morphology, 4 patients underwent biventricular endocardial 3D-EAM, while 23 patients underwent only LV endocardial 3D-EAM, and one patient only RV endocardial 3D-EAM. In the non-CTO group, 26 patients underwent biventricular endocardial 3D-EAM; 41 and 25 patients underwent only LV and RV endocardial 3D-EAM, respectively. Epicardial mapping was performed in 7 patients (6% of the entire population; 2 in the CTO group and 5 in the non-CTO group). In the CTO group, the proportion of LV VA exit guided by LV endocardial 3D-EAM (96% vs. 73%, *p* < 0.001) and the prevalence of LAVA were greater than in the non-CTO group (71% vs. 25%, *p* < 0.001), without difference among vessels responsible for CTO (Table 3, Figure 3). The number and the extent of scar areas were similar between the 2 groups. In the subset of ischemic patients (43 patients), LV scar areas were more frequently detected by 3D-EAM in the CTO group than in the non-CTO group (35% versus 6%, *p* = 0.01).

The acute success rate (in terms of non-VAs inducibility plus complete LAVA abolition plus acute PVC suppression) in the two groups was similar (23/28 pts, 83% in the CTO group vs. 76/92 pts, 82% in the non-CTO group, *p* = 0.434). 

### 3.3. Follow-Up Findings

VAs recurrence was higher in the CTO group (18/28, 64%) compared with non-CTO group (43/92, 46%) (*p* = 0.03) after a mean follow-up of 20 ± 3.6 months (median follow up 15 months, range 1–96). During this period, none of the CTO patients underwent coronary revascularization. With regard to the use of beta-blockers and amiodarone during the follow-up period, we did not find a difference between the CTO and non-CTO groups, respectively amiodarone in non-CTO group 26% vs. CTO group 30%, beta-blockers 45% in non-CTO group vs. 55% in CTO group. At univariate Cox regression analysis, the presence of CTO and LV systolic dysfunction were predictors of VAs recurrence (HR 1.8, 95% CI 1.03–3.1, *p* = 0.04 and HR 2.6, 95% CI 1.5–4.6, *p* = 0.001, respectively) (Table 4). At multivariate analysis, the only independent predictor of VAs recurrence was LV dysfunction (LVEF < 35%) (*p* = 0.02) (Table 4, Figure 4).

## 4. Discussion

The main finding of our single-centre observational study in patients undergoing catheter ablation for VAs is the higher recurrence rate of VAs in the CTO group as compared with the non-CTO group. Nevertheless, the presence of unrevascularized CTO is not an independent predictor in patients with VAs undergoing catheter ablation and LVEF remains the only statistically significant predictor of VAs recurrence at mid-term follow-up. 

CTO is a common condition affecting patients with ischemic heart disease and involves a sizable percentage (67%) of ischemic patients referred to the cath lab. Among these patients, there is a higher rate of ICD carriers, with the device being implanted both in primary and secondary prevention and CTO seems to confer a higher risk of VA recurrence in mid-long-term follow-up [3,4,11]. However, data on CTO as independent predictors of VA recurrence after catheter ablation are still scarce. 

To the best of our knowledge, there is only one single-centre observational study that identified IRA-CTO as a predictor of VAs recurrence in a population submitted to catheter ablation in which the acute success was defined as LAVA abolition plus no VAs reinducibility [5]. In addition, Di Marco et al. demonstrated that the combination of IRA-CTO and the absence of LAVA abolition during ablation was the strongest predictor of VT recurrence. In contrast, IRA-CTO lost its predictive role for VAs recurrence when combined with successful LAVA abolition. 

In accordance with these results, the achievement of the complete endpoint (LAVA abolition plus non-inducibility plus acute PVC suppression) was 83% in our population, and it seems to have a favourable impact on VAs recurrence in patients with VAs and CTO at 1-year of follow up. Consequently, the electroanatomic substrate of patients with CTO could represent a potentially modifiable factor by radiofrequency catheter ablation. 

Several reports suggest that patients with CTO and especially with IRA-CTO not undergoing ablation are at high risk for VAs due to scar-related re-entry circuits [3,4,5,6]. In accordance with Di Marco et al., in our study patients with CTO had more LV scar areas and a higher prevalence of LAVA compared with the non-CTO group, with consequent more arrhythmogenic substrate and need for a complete substrate elimination as achieved by LAVA abolition. 

Of note, we found that patients with LAD CTO (all IRA CTO) compared with a non-ischemic and heterogeneous population of patients undergoing VAs catheter ablation presented with a higher risk of VAs recurrence at the follow-up. Yet CTO failed to represent an independent predictor of VAs at variance with previous studies [4,5,6]. Particularly, in VACTO Secondary study, patients with CTO were more likely to receive an appropriate ICD therapy compared with those without CTO; moreover, the presence of CTO and lower LVEF at baseline were independent predictors of VAs recurrence. However, this multicenter study enrolled ischemic patients receiving an ICD for secondary prevention and not undergoing VAs catheter ablation. Moreover, in a large cohort of primary prevention ICD recipients, the presence of a CTO was not associated with a higher incidence of VAs [12]. To the best of our knowledge, regardless of the underlying heart disease, severe ventricular dysfunction (LVEF < 35%) remains the only independent variable to identify patients at high risk of VAs after catheter ablation [13]. Accordingly, in our study the only independent predictor of VAs recurrence at multivariate analysis was the occurrence of moderate to severe LV dysfunction; the presence of CTO does not seem to give an incremental risk of VA recurrences, after catheter ablation. Yap et al. found a similar survival rate between the CTO group without LV dysfunction and the non-CTO group with severe LV dysfunction, in patients with aborted SCD not undergoing ablation; thus, the presence of CTO contributed to arrhythmic risk equally with severe LV dysfunction in patients not undergoing ablation [14].

There is growing interest in percutaneous treatment of CTO due to observational evidence suggesting that CTO revascularization could improve LV function and survival [15,16], but it is unknown whether this might have an impact on ventricular arrhythmogenesis. In our study, we found a lack of association between presence of CTO and VAs recurrence after catheter ablation, suggesting that a strategy aiming at careful LAVA mapping and complete ablation on top of non reinducibility could be a valuable strategy to reduce the arrhythmic risk in these patients. Further studies are needed to evaluate which strategy is more effective to reduce VA in CTO patients.

Several limitations should be acknowledged. This observational study is limited by the low number of patients, with particular regard to IRA-CTO patients. Therefore, our results should be confirmed in larger populations. As a single-center study, biases related to selection criteria and local population may have influenced the results.

Moreover, the heterogeneity of the non-CTO group could have affected the results. Indeed, the CTO group is a high-risk population, while the non-CTO group includes an intermediate-risk population, including 27% of patients with idiopathic outflow tract VT. It could explain the fact that in our study there is a difference in VAs recurrence between the two groups, but CTO is not an independent predictor of VAs recurrence, after catheter ablation.

In conclusion, we found that among patients undergoing catheter ablation for ventricular arrhythmias LV dysfunction is the only independent predictor of VA recurrence, regardless of baseline cardiomyopathy, while CTO predicts VA recurrence at univariate but not at multivariate analysis.

## Figures and Tables

**Figure 1 jcm-11-07181-f001:**
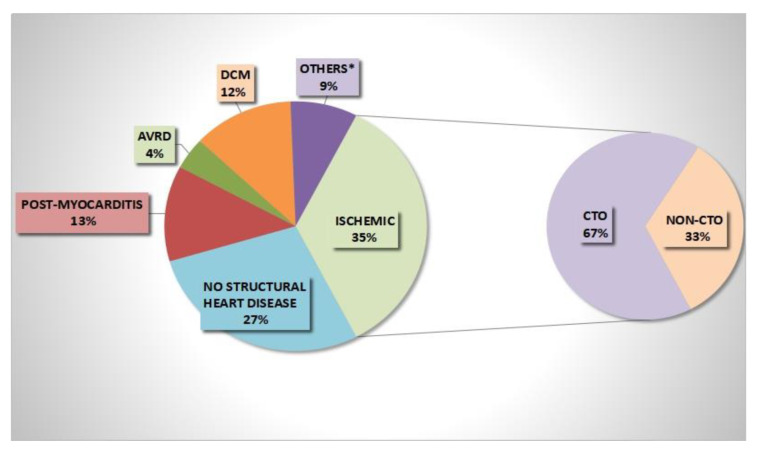
Underlying heart diseases on admission. Others *: hypertrophic cardiomyopathy, grown-up congenital heart disease, valvular heart disease.

**Figure 2 jcm-11-07181-f002:**
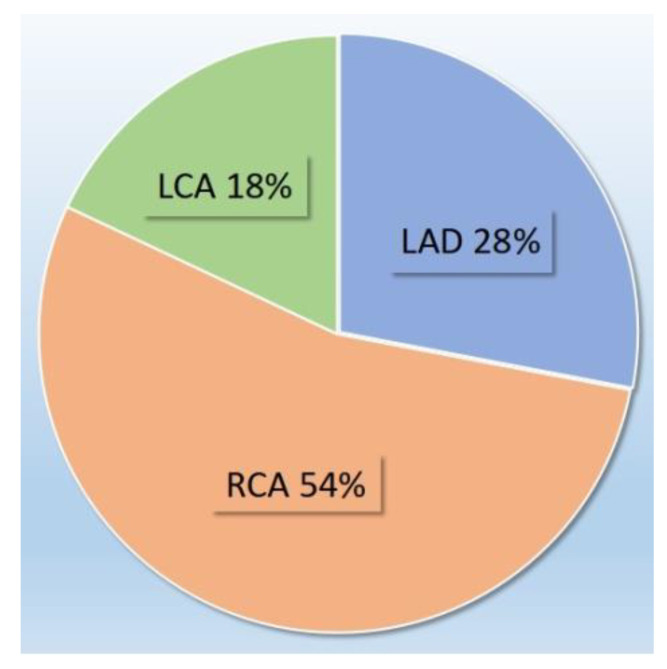
Coronary artery CTO distribution.

**Figure 3 jcm-11-07181-f003:**
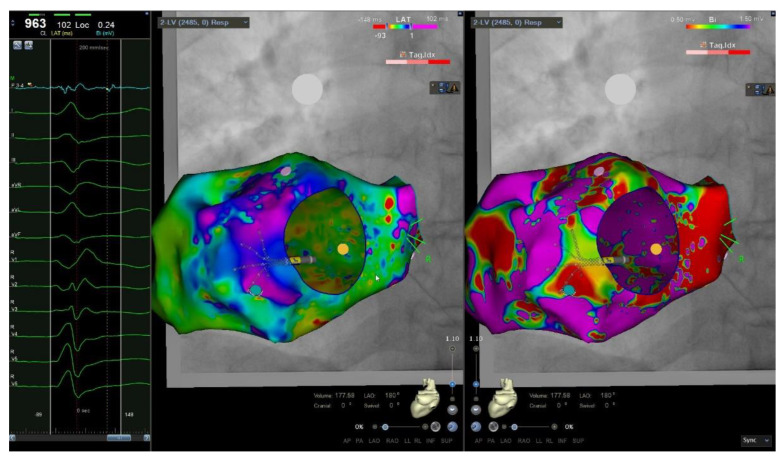
Local abnormal ventricular activities and substrate endocardial 3D-EAM of left ventricle in patient with CTO. (**Left Panel**): left ventricular endocardial electrograms during 3D-EAM: local abnormal ventricular activities recorded after QRS during left ventricular 3D-EAM in sinus rhythm of patient with LCA-CTO. (**Central Panel**): activation mapping during sinus rhythm of patient with LCA-CTO; the green dot is the site of recorded LAVA in lateral basal left ventricular wall. (**Right Panel**): bipolar endocardial 3D-EAM of left ventricle of patient with LCA-CTO with evidence of dense scar (red zone with potentials < 0.5mV) and border zone (green and blu zones with potentials ranging 0.5–1.5 mV) in lateral basal left ventricular wall.

**Figure 4 jcm-11-07181-f004:**
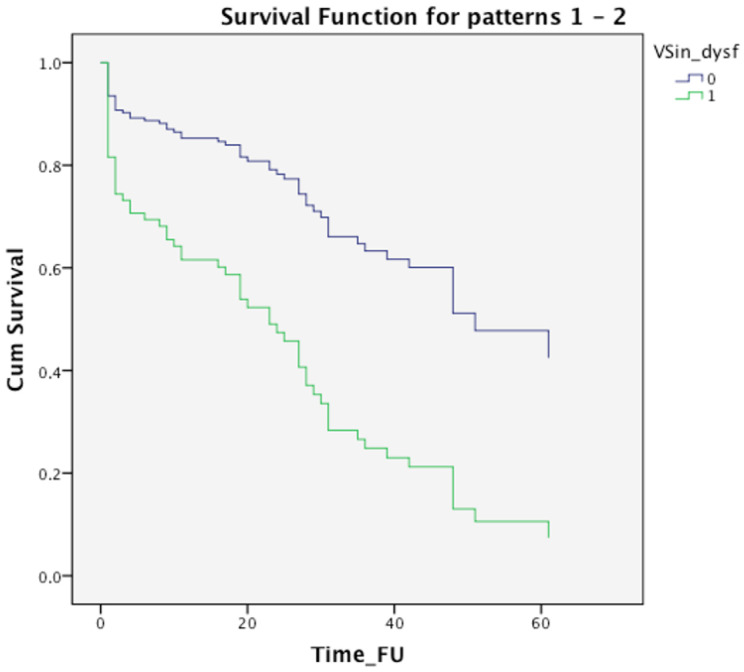
Kaplan-Meier survival curves for VAs recurrence according to Left ventricular Ejection Fraction (LVEF) ≤35% or >35%.

**Table 1 jcm-11-07181-t001:** Baseline clinical features on admission.

Clinical Data	No CTO pts (92)	CTO pts (28)	*p* Value
Men, n (%)	63 (68%)	27 (96%)	0.002
Age, y, mean ± SD	56 ± 16	68 ± 8	<0.001
Hypertension, n (%)	44 (47%)	24 (86%)	<0.001
Diabetes mellitus, n (%)	24 (26%)	9 (32%)	0.37
Smoke, n (%)	13 (14%)	0 (0)	0.07
COPD/OSAS, n (%)	16 (17%)	6 (21%)	0.27
Previous MI, n (%)	10 (11%)	27 (96%)	<0.001
Sustained VAs on admission	49 (53%)	27 (96%)	<0.001
Chronic kidney disease, n (%)	25 (27%)	16 (57%)	0.005
NYHA class, n (%)	32 (35%)	18 (64%)	0.06
ICD carriers, n (%)	41 (44%)	23 (82%)	0.005
Electrical storm, n (%)	29 (31%)	11 (39%)	0.14
Atrial fibrillation, n (%)	31 (34%)	11 (39%)	0.15
AAD on admission, n (%)			
B-blockers	36 (39%)	15 (53%)	0.09
Amiodarone	17 (19%)	12 (42%)	0.003

Table legend: CTO: Coronary total occlusion; COPD: chronic obstructive pulmonary disease; OSAS: Obstructive Sleep Apnea Syndrome; MI: myocardial infarction; VAs: ventricular arrhythmias; ICD: implantable cardioverter defibrillator; AAD: anti-arrhythmic drugs.

**Table 2 jcm-11-07181-t002:** Baseline echocardiographic findings on admission.

Echocardiography Data	No CTO pts (92)	CTO pts (28)	*p* Value
LVEF, mean, % ± SD	48 ± 13	36 ± 10	<0.001
Moderate/Severe LV dysfunction LVEF < 40%, n (%)	28 (30%)	18 (64%)	0.006
RV dysfunction (TAPSE < 17 mm), n (%)	15 (16%)	3 (11%)	0.81
Akinetic ventricular areas, n (%)	20 (22%)	21 (75%)	<0.001

Table legend: CTO: Coronary total occlusion; LVEF: left ventricular ejection fraction; TAPSE: tricuspid annular plane excursion.

**Table 3 jcm-11-07181-t003:** Left and right ventricular 3D-Electroanatomic mapping findings.

EAM Data	No CTO pts (92)	CTO pts (28)	*p* Value
LV Endocardial 3D-EAM, n (%)	67 (73%)	27 (96%)	<0.001
Total volume mL, n ± SD	214 ± 101	212 ± 68	0.864
Unipolar scar, n (%)	33 (36%)	14 (50%)	0.117
Unipolar scar area, cm^2^, mean ± SD	99 ± 96	85 ± 68	0.457
Bipolar scar, n pts (%)	21 (23%)	11 (39%)	0.058
Bipolar scar area, cm^2^, mean ± SD	36 ± 40	33 ± 28	0.992
LV VA exit by 3D-EAM, n (%)	51 (55%)	24 (86%)	0.001
RV Endocardial 3D-EAM, n (%)	51 (55%)	5 (18%)	<0.001
Total volume, mL, n ± SD	105 ± 40	130 ± 2	0.839
Unipolar scar, n (%)	12 (13%)	0	-
Unipolar scar area, cm^2^, mean ± SD	27 ± 26	0	-
Bipolar scar, n (%)	6 (6.5%)	0	-
Bipolar scar area, cm^2^, mean ± SD	20 ± 20	0	-
Endocardial LAVA, n (%)	23 (25%)	20 (71%)	<0.001
Epicardial 3D-EAM, n (%)	7 (7%)	2 (7%)	0.786
Bipolar scar, n (%)	6 (6%)	2 (7%)	0.571
Bipolar scar area, cm^2^, ± SD	105 ± 42	119 ± 3	0.593
Epicardial LAVA n (%)	5 (5%)	2 (7%)	0.391

Table legend: CTO: Coronary total occlusion; 3D-EAM: three-dimension electroanatomic mapping; LAVA: local abnormal ventricular activities; LV left ventricular; VA ventricular arrhythmias.

**Table 4 jcm-11-07181-t004:** Univariate and Multivariate Cox regression analysis.

Variables	HR (95% CI)	*p* Value	HR (95% CI)	*p* Value
Age	1 (0.9–1.03)	0.5		
Sex	0.9 (0.5–1.8)	0.9		
Hypertension	1.4 (0.8–2.4)	0.1		
Type II DM	0.7 (0.4–1.4)	0.3		
Previous MI	1.5 (0.9–2.5)	0.1		
Presence of CTO	1.8 (1.03–3.1)	0.04	1.5 (0.9–2.5)	0.13
IRA CTO	1.6 (0.53–1.9)	0.4		
Number of CTO	1.2 (0.83–1.8)	0.2		
LAD CTO	2.1 (0.9–5)	0.07		
LCA CTO	0.9 (0.2–3.8)	0.9		
RCA CTO	1.5 (0.8–2.8)	0.2		
LVEF < 35%	2.6 (1.5–4.6)	0.001	3.1 (1.5–6.4)	0.002
Akinetic areas	0.9 (0.4–1.7)	0.7		
NYHA class > II	1.3 (1.05–1.7)	0.02		
ICD carriers	1.7 (1.3–2.4)	0.001		
Endocardial LAVA/LP	4 (1.8–9)	0.001		
Beta-blockers at FUP	0.65 (0.2–1.6)	0.37		
Amiodarone at FUP	0.87 (0.5–1.4)	0.59		

Table legend: DM: diabetes mellitus; MI: myocardial infarction; CTO: chronic total occlusion; ICD: implantable cardioverter defibrillator; IRA: infarct-related artery; LAD: left anterior descending artery; LCA: left circumflex artery; LAVA: local abnormal ventricular activities; LP: late potentials; LVEF: left ventricular ejection fraction; MI: myocardial infarction; RCA: right coronary artery; LVEF: left ventricular ejection fraction; NYHA: New York Heart Association; FUP: follow-up.

## Data Availability

The data presented in this study are available on request from the corresponding author.

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
