# Peer review of "Mid-Term Outcome of Ventricular Arrhythmias Catheter Ablation in Patients with Chronic Coronary Total Occlusion Compared to Ischemic and Non-Ischemic Patients"

_jcm, 2022, doi:10.3390/jcm11237181_

Round 1

Reviewer 1 Report

The authors' present the long-term outcomes of patients who underwent catheter ablation of ventricular tachycardias (VTs). Although the data on outcomes in  VT ablation procedures is scarce, it is difficult to extract much novelty from the present study.

Major issues:

1. The title of the article is somehow misleading as only 23% of the patients had CTO, while many patients did not even have ischemic cardiomyopathy.

2. Definition of complex ventricular arrhythmias is difficult to comprehend as premature complexes and non-sustained VTs can not be considered complex, although some procedures to abolish these PVCs can be challanging.

3. To assess the outcomes after VA ablation patients with similar structural diseases should be compared, therefore I would suggest that only patients with ischemic cardiomyopathy should be assessed; comparison of CTO and nonCTO patients with ischemic cardiomyopathy.

4. What was the definition of the electrical storm? It is very odd that ES were so frequent in nonCTO group, since the population has less comorbidites, better EF and had less previous MI?

5. Baseline characteristics in Table 1 are difficult to compare as both non-CTO and CTO populations could not be compared. Instead, only baseline characteristics of ischemic patients with CTO and nonCTO should be maybe considered. 

6. IRA-CTO patients are more prone to VAs compared to nonCTO and non-IRA CTO patients, therefore some additional imaging data on this patients would be helpful to further analyse the effect of VT ablation in these patients.

Minor issuses:

1. VAs should written be in capital letters.

Author Response

The authors' present the long-term outcomes of patients who underwent catheter ablation of ventricular tachycardias (VTs). Although the data on outcomes in  VT ablation procedures is scarce, it is difficult to extract much novelty from the present study.

We thank the Reviewer for their careful review, which helped improve the manuscript. Please find our answer to comments in blue as well as suggested text changes in yellow.

Major issues:

  1. The title of the article is somehow misleading as only 23% of the patients had CTO, while many patients did not even have ischemic cardiomyopathy.

The purpose of our study is to evaluate  if CTO is an independent predictor of VAs recurrence after ablation and for this reason we compared VA recurrence in CTO group vs non CTO group (including ischemic and non ischemic patients). Therefore, in accordance with Reviewer, we propose another title: “Long-term outcome of ventricular arrhythmias catheter ablation in patients with chronic coronary total occlusion compared to ischemic and non-ischemic patients“.

  1. Definition of complex ventricular arrhythmias is difficult to comprehend as premature complexes and non-sustained VTs can not be considered complex, although some procedures to abolish these PVCs can be challanging.

In accordance with Reviewer,  we replaced  the term “complex VAs” with “VAs”  in the new version of manuscript.

  1. To assess the outcomes after VA ablation, patients with similar structural diseases should be compared, therefore I would suggest that only patients with ischemic cardiomyopathy should be assessed; comparison of CTO and nonCTO patients with ischemic cardiomyopathy.

In order to evaluate if CTO is an independent predictor of VAs recurrence after catheter ablation, we initially evaluated the difference in VAs recurrence among CTO  and non-CTO patients (ischemic and non ischemic patients). Subsequently, in the subgroup of ischemic patients (43 patients, 28 with diagnosis of CTO and 15 without CTO), we reported in detail clinical and diagnostic characteristics in Result Section.

  1. What was the definition of the electrical storm? It is very odd that ES were so frequent in nonCTO group, since the population has less comorbidites, better EF and had less previous MI?

We defined electrical storm (ES) as“3 or more sustained episodes of ventricular tachycardia, ventricular fibrillation, or appropriate shocks from an implantable cardioverter-defibrillator within 24 hours.” (From “The Evaluation and Management of Electrical Storm” Michael Eifling, MD, Mehdi Razavi, MD, and Ali Massumi, MD).

According to this definition, we found 31% of ES in non CTO group in and 39% of ES in CTO group.

  1. Baseline characteristics in Table 1 are difficult to compare as both non-CTO and CTO populations could not be compared. Instead, only baseline characteristics of ischemic patients with CTO and non CTO should be maybe considered.

In Table 1  we reported baseline characteristics of the two main groups (CTO vs non-CTO groups); in the Results Section, Baseline findings, we reported in detail the characteristic of ischemic groups (43 patients).

  1. IRA-CTO patients are more prone to VAs compared to non CTO and non-IRA CTO patients, therefore some additional imaging data on this patients would be helpful to further analyse the effect of VT ablation in these patients.

We added in the new version of manuscript supplemental figure 5 with evidence of late potentials activity in left ventricular endocardial substrate of right coronary artery occlusion.

Minor issuses:

1.VAs should written be in capital letters.

We replaced VAs in accordance with Reviewer.

Reviewer 2 Report

The authors assessed the effect of CTO on complex VA occurence and the only  statistically significant finding was the presence of LV systolic dysfunction. 

Low number of relevant patients is the most important limitation. On the other hand, there is no data regarding the beta blocker or other AAD usage during the 20-month follow-up period. The authors should include data about this point in the text. The same concern should be addressed as regards of the LVEF.

At baseline, only half of the patients receive beta blocking agents despite most of them have ICDs. The authors shoud discuss this issue in the text.

Author Response

The authors assessed the effect of CTO on complex VA occurence and the only  statistically significant finding was the presence of LV systolic dysfunction.

We thank the Reviewer for their positive comment and careful review, which helped to improve the manuscript. Please find our answers to the comments in blue as well as suggested text changes in yellow.

Low number of relevant patients is the most important limitation. On the other hand, there is no data regarding the beta blocker or other AAD usage during the 20-month follow-up period. The authors should include data about this point in the text. The same concern should be addressed as regards of the LVEF.

With regard to the use of beta-blockers and amiodarone during the follow-up period, we did not find difference between CTO and non CTO groups, respectively amiodarone in  non-CTO group 26% vs CTO group 30%, betablockers 45% in non-CTO group vs 55% in CTO group.  We Added this sentence in Results Section: Follow up findings (page 7, lines 230-233).

At baseline, only half of the patients receive beta blocking agents despite most of them have ICDs. The authors shoud discuss this issue in the text.

In one third of patients beta-blockers and other antiarrhythmic drugs were discontinued before the electrophysiological study in order to facilitate VAs induction during the procedure. This sentence was added in the new version of the manuscript: Materials and Methods Section, Page 2 and line 79-81.

Round 2

Reviewer 2 Report

please add beta blocker and amiodarone usage to the regression analysis and discuss the relevant results in discussion section in an attempt to improve the yield of the article

Author Response

With regard to the use of beta-blockers and amiodarone during the follow-up period, we did not find difference between CTO and non CTO groups, respectively amiodarone in  non-CTO group 26% vs CTO group 30%, betablockers 45% in non-CTO group vs 55% in CTO group.  We Added this sentence in Results Section: Follow up findings (page 7, lines 230-233).

Regarding the second revision,  we added in revised table n. 4 (univariate analysis) the HR (95% IC) and p value related to Amiodarone and Betablockers at follow-up.